# Porcine Small Intestinal Submucosa Alters the Biochemical Properties of Wound Healing: A Narrative Review

**DOI:** 10.3390/biomedicines10092213

**Published:** 2022-09-07

**Authors:** Miki Fujii, Rica Tanaka

**Affiliations:** 1Division of Regenerative Therapy, Juntendo University Graduate School of Medicine, Tokyo 113-8421, Japan; 2Department of Plastic and Reconstructive Surgery, Juntendo University School of Medicine, Tokyo 113-8421, Japan

**Keywords:** wound healing, extracellular matrix, small intestinal submucosa, dynamic reciprocity, M2-macrophage, matrix metallic proteinases, matrix-bound vesicle

## Abstract

Among the many biological scaffold materials currently available for clinical use, the small intestinal submucosa (SIS) is an effective material for wound healing. SIS contains numerous active forms of extracellular matrix that support angiogenesis, cell migration, and proliferation, providing growth factors involved in signaling for tissue formation and assisting wound healing. SIS not only serves as a bioscaffold for cell migration and differentiation, but also restores the impaired dynamic reciprocity between cells and the extracellular matrix, ultimately driving wound healing. Here, we review the evidence on how SIS can shift the biochemical balance in a wound from chronic to an acute state.

## 1. Introduction

The wound healing process comprises distinct but often overlapping stages, namely hemostasis, inflammation, proliferation, and maturation. To proceed to the next step of wound healing, interactions between cells and their microenvironment must occur, in which the extracellular matrix (ECM) is the primary component. Cells secrete the ECM that surrounds cells in tissues. The ECM has long been understood to structurally support cells, since its characteristics set the characteristics of the tissue (i.e., bone compared to cartilage compared to brain) [1]. However, instead of being a passive mechanical support for cells, it is an extraordinarily complex scaffold composed of a variety of highly regulated biologically active molecules that are critical to determine the action and fate of cells [2]. Because the interactions between cells and ECM are reciprocal, dynamic, and constantly changing, the interactions are known as dynamic reciprocity (DR) (Figure 1) [3,4].

Acute and chronic wounds have vastly different wound environments, with excessive inflammation and abnormalities in the ECM of the chronic wound that adversely affect wound healing [5]. Normal wound healing is triggered by tissue injury. Vascular damage leads to extravasation of blood components and exposure of ECM proteins, such as collagen, fibronectin, laminin, and matricellular protein thrombospondin. As blood components spill into the injury site, platelets come into contact with exposed collagen and other elements of the ECM. This binding triggers platelets to degranulate and release clotting factors, as well as essential growth factors and cytokines such as platelet-derived growth factor (PDGF) and transforming growth factor-beta (TGF-β), which modulate cell proliferation and matrix metabolism. Following hemostasis, neutrophils enter the wound site and begin phagocytosis to remove foreign materials, bacteria, and damaged tissue. Subsequently, macrophages appear and continue the phagocytic process, releasing additional PDGF and TGF-β. During the inflammation phase, the wound bed is set for healing, and at the same time, fibroblasts are recruited and activated. Once the wound site is cleaned out, fibroblasts migrate to the injury site, beginning the proliferative phase and depositing new ECM. The new collagen matrix becomes cross-linked and organized during the final remodeling phase [6].

Chronic wounds exhibit a prolonged inflammatory response. In acute wounds, inflammation is a self-limiting process. In contrast, inflammation causes more injury in chronic wounds, further fueling additional inflammation in a detrimental cycle. One potential cause is excessive neutrophil infiltration [7]. Although activated neutrophils are virtually nonexistent after the first 72 h in acute wounds, neutrophils are present throughout the healing process in chronic wounds. Possible reasons for neutrophil persistence include continued recruitment and activation due to tissue trauma caused by pressure, bacterial overgrowth, leukocyte trapping, and ischemic-reperfusion injury. The large number of activated neutrophils leads to excessive amounts of degradative matrix metallic proteinases (MMPs), especially MMP-8 and neutrophil-derived elastase. In a normal wound, all MMPs can be inhibited by the nonspecific proteinase inhibitor, α2-macroglobulin, and specifically by a small group of proteins called tissue inhibitors of matrix metalloproteinases (TIMPs). However, in nonhealing wounds, MMPs are not balanced by equal TIMPs. Consequently, an abnormal degradative/protective enzyme ratio ensues, which favors wound degradation. The excessive abundance of inflammatory cells also affects the cytokine profile in the wound, where inflammatory cytokines predominate, including tumor necrosis factor-α (TNF-α). Furthermore, factors that promote proliferation and matrix deposition, such as PDGF and TGF-β, respectively, are scarce. Neutrophils are activated, resulting in enzyme release, tissue degradation, and further neutrophil recruitment, creating a vicious cycle. Therefore, fibroblasts cannot deposit ECM, since collagen degradation outpaces its synthesis. Therefore, an uncontrolled inflammatory response prevents rather than promotes wound healing. Not only the inflammatory profile is altered, but fibroblasts in a chronic wound are also altered.

From the perspective of DR, chronic wounds lose the typical sequence of reactions between cells and the ECM that characterizes wound healing. Replacing dysfunctional ECM is one method to improve DR. Porcine small intestinal submucosa (SIS) is a thin, translucent layer and a naturally occurring ECM derived from the submucosal layers of the porcine jejunum. The porcine small intestine is subjected to manufacturing processes to isolate and clean the submucosa that frees the material from viral contamination and minimizes the risk of transmission of zoonotic diseases [8]. SIS is freeze-dried and sterilized using ethylene oxide gas to prepare for clinical use [8]. It contains biologically important components of the ECM, such as several collagens (types I, III, IV, and VI), glycosaminoglycans, proteoglycans, and fibronectin, as well as growth factors, including basic fibroblast growth factor (FGF-2), vascular endothelial cell growth factor (VEGF), and TGF-β. Among the many biological scaffold materials currently available for clinical use, SIS is effective for wound healing. The components of SIS are retained in their active forms; therefore, they can fuel angiogenesis, cell migration, and proliferation, providing growth factors to facilitate signaling for tissue formation and aid wound healing [9]. SIS is useful as a bioscaffold for cell migration and differentiation, as well as for repairing the altered DR between cells and the ECM, which ultimately leads to wound healing. In this review, we summarize the evidence regarding how SIS can shift the biochemical balance in a wound from a chronic to an acute state by improving DR, leading to wound healing.

## 2. SIS Promotes Constructive Remodeling through DR

There are many medical devices composed of allogeneic and xenogeneic ECM. However, the range of performance varies depending on the source of the material, the preparation methods, and clinical application [10]. The ideal ECM-based product should promote constructive remodeling, which is the de novo formation of a site-appropriate functional tissue [11]. Woo et al. [12] showed the effect of SIS on morphological characteristics and biochemical components during the healing of medial collateral ligaments (MCL) in rabbit models. Twelve weeks after SIS treatment, the orientation of cells and ECM in the SIS-treated group was much more organized along the longitudinal direction of the healing ligament, with a concomitant 22.2% increase in collagen fibril diameters over the nontreated group. Furthermore, a reduction in type V collagen was observed, with a 28.4% reduction in the collagen type V/I ratio using immunofluorescent staining. Salgado et al. [13] showed the quality of epithelialization using burns of mid-partial thickness in human patients. Five patients with burns affecting up to 10% of the body’s surface were treated with SIS and a silver-containing hydro fiber (AgH) in parallel, but in different areas. The tissues treated with SIS were significantly more phenotypically structured after seven days. A higher epithelial maturation index (6.2 ± 0.84 vs. 3.2 ± 3.28; *p*
*=* 0.029), better orientation and differentiation of epithelial cells, as well as an appropriate basal lamina structure, collagen deposition, and higher TNF-β3 expression in the dermis (7.4 ± 8.1 vs. 2.1 ± 2.6; *p* = 0.055) were observed in tissues treated with SIS compared to those with AgH dressings. After three months of treatment, SIS treatment resulted in a lower Vancouver Scar Scale score, especially in vascularity, pigmentation, and pliability (3.6 ± 2.6 vs. 7.2 ± 2.5, *p* = 0.025).

Devices made from SIS are commonly used for surgical repair or reconstruction and have exhibited constructive remodeling. Nobuyma et al. [14] performed a constructive remodeling with a tendon-exposed wound on the dorsal aspect of the foot. The surface of the tendon is poorly vascularized and relies on synovial fluid from the synovial tendon sheath for nourishment. However, if the sheath is absent, the blood flow on the tendon surface is poor, the granulation tissue does not form, and the skin graft is rejected. SIS successfully promoted cell proliferation from the surrounding cells to the interior, and a dermis-like tissue was formed on the tendon, resulting in the formation of a good scaffold. Furthermore, the ability of the foot to dorsiflect was maintained without adhesion or scar contracture post operation. Badylak et al. [15] described the placement of SIS after the removal of the esophageal mucosa and submucosa in patients with Barrett’s esophagus. The five patients all experienced the restoration of normal, mature squamous epithelium, and a normal diet. Bejjani et al. [16] reported the results of a prospective multicenter study that evaluated the use of SIS as a dural repair material. They reported that in all 59 patients, after a mean follow-up of 7.3 ± 2.2 months, the SIS demonstrated substantial clinical efficacy. These studies indicate SIS does not only act as scaffolding for the wound, but it also promotes a functional, constructive remodeling.

Diabetic foot ulcers (DFUs) are one of the leading causes of morbidity, economic loss, and decreased quality of life in affected patients. For the past several years, biomaterials derived from natural tissue sources have been used to stimulate wound closure. In a perspective, randomized clinical trial reported by Niezgoda et al. [17], 73 patients with DFUs were randomly treated with standard care and SIS (*n* = 37) and compared to patients treated with standard care and becaplermin (*n* = 36). More wounds in the SIS-treated group healed at 12 weeks (49% vs. 28%; *p* = 0.055). Although not statistically superior to PDGF treatment, SIS may be a viable treatment option for chronic DFU. Based on these observations, SIS has been recommended as an adjunctive therapy for DFUs, when recalcitrant to standard therapy (Grade 2C) according to the clinical guidelines of the USA [18].

In addition, venous leg ulcers are another major cause of refractory ulcers. SIS was recommended to patients who did not show signs of healing after standard therapy according to the guidelines, including compression for 4 to 6 weeks [19]. Prospective data from the randomized clinical trial (RCT) [20] showed the efficacy of the compression-treated SIS wound matrix compared to compression alone in the healing of chronic leg ulcers. After 12 weeks of treatment, 55% of the wounds in the SIS group healed, compared to 34% of those in the standard care group (*p* = 0.0196).

## 3. Reducing Elevated Levels of MMPs

MMPs are a family of zinc endopeptidases responsible for ECM degradation, which is essential for cell migration and wound healing. In acute wounds, fibroblast production and activation of MMPs are regulated by numerous factors, such as mechanical strain, growth factors, and cell interactions with the ECM. Typically, wounds heal as expected. However, extensive damage to the ECM can disrupt cell function and migration, leading to the collapse of the DR between cells and the ECM and ultimately impair the wound healing process. The large number of activated neutrophils leads to excessive amounts of MMP, which may be an underlying reason for the prolonged inflammatory phase. Many previous studies revealed elevated MMP levels in chronic wounds and a correlation between elevated MMP levels and non-healed wounds [3,21]. Therefore, reducing the high levels of MMP in chronic wounds is key to promoting healing. Hodde et al. [22] investigated the serum and wound exudates of patients with chronic venous leg ulcers treated with SIS. These exudates collected from seven patients whose wounds healed after SIS treatment showed significant decreases in MMP-1, MMP-2, MMP-3, MMP-9, TNF-α, and interleukin (IL)-8 (*p <* 0.05), as well as substantial increases in TGF-β1 in the wounds, at 12 weeks. However, MMP and TGF-β levels remained elevated and depressed, respectively, in five patients who did not respond to treatment. None of the 12 patients showed a measurable serum antibody response to SIS. These results indicate that the treatment of chronic venous ulcers with SIS resets the wound environment to an acute biochemical state, allowing the wounds to heal. Yeh et al. [23] compared the pathogenesis of skin biopsies of acute wounds treated with negative pressure wound therapy (NPWT) alone and combined with SIS. Wounds treated with NPWT and SIS showed a significant decrease in inflammation scores in histological analysis. In addition, SIS-treated wounds showed more advanced wound healing and greater wound contraction. Thus, using SIS in wounds should be considered a method to reduce inflammation and accelerate wound healing at clinical and histological levels compared to using NPWT alone.

Excessive neutrophil activation also leads to an increased production of neutrophil-derived elastase (HNE), which degrades connective tissue molecules and activates MMPs [24]. A recent publication studying the proteomic profile of ECM reported that SIS contains leukocyte elastase inhibitor [25]. This suggests that SIS may have the potential to reduce elevated levels of HNE.

## 4. From Inflammation to Proliferation

In the early stages of wound repair, monocytes differentiate into the M1 subset of macrophages, which are associated with phagocytic activity, scavenging, and the production of proinflammatory mediators (Figure 2) [26]. The M1 macrophages are transformed into the M2 subset, exhibiting a reparative phenotype. M2 macrophages help synthesize anti-inflammatory mediators and produce ECM during the initiation of fibroblast proliferation and angiogenesis [26,27,28] (Figure 3). Macrophages play a key role in the transition from the inflammatory to the proliferative stage of wound repair. If the M1–M2 transition does not occur, it can cause nonhealing or chronic wounds, such as venous ulcers and diabetic wounds. Several studies have shown that an appropriately timed transition in macrophage activation state is necessary to promote tissue remodeling and wound healing rather than scar tissue formation at various anatomic sites, including the skeletal muscle [29] and myocardium [30].

Some reports have indicated that SIS facilitates the M1–M2 transition and promotes wound healing to advance from inflammation to proliferation. Dziki et al. [31] investigated the mechanisms of SIS treatment in volumetric muscle loss (VML) injuries. VML is a massive or excessive skeletal muscle injury resulting from trauma, tumor ablation, or degenerative disease. Skeletal muscle can repair in response to mild trauma. However, the skeletal muscle damage associated with VML overwhelms the regenerative process, resulting in scar tissue deposition, loss of function, and esthetic deformities. The therapeutic options for massive loss of skeletal muscle tissue after trauma, surgical excision of neoplasms, and related conditions are limited. Autologous muscle grafts or muscle transposition represent optional salvage procedures for the restoration of muscle tissues; however, these approaches are limited by the availability of donor tissue. Alternatively, cell-based therapies have been explored, but problems still arise in ES/iPS cell therapies, such as immunogenicity, the desirable requirement of the source, and a favorable environment to maintain cell viability [4]. Furthermore, the development of a therapy that avoids the collection, isolation, and/or ex vivo expansion and purification of autologous stem cells with subsequent reintroduction to the patient would reduce the regulatory hurdles for clinical translation, reduce the cost of treatment, and avoid the risks associated with cell-based approaches. There is an unmet need for therapeutic strategies that can enhance the innate regenerative capacity of skeletal muscle following VML [31]. Dziki et al. [31] analyzed the spatiotemporal patterns of myogenic and neurogenic progenitor cells and the pattern of macrophage activation following SIS in a murine model of VML. The group treated with ECM showed an early transition from predominant M1-like macrophages to robust M2-like macrophages. SIS treatment promoted perivascular stem cell mobilization, increased the presence of neurogenic progenitor cells, and was associated with myotube formation. These cell types were present not only at the periphery of the defect near the uninjured muscle, but also at the center of the SIS-treated defect. Furthermore, SIS altered the wound microenvironment, allowing synergy and crosstalk between key immune regulators and skeletal muscle progenitor cells, whose spatiotemporal distribution is associated with de novo skeletal muscle formation and site-appropriate remodeling. Based on these results [29], SIS can induce site-specific remodeling in the organs or tissues where it is placed. Although SIS is a xenogeneic substance, clinical or histological evidence of immediate or delayed rejection has not been reported. Allman et al. [32] performed histological analysis to detect rejection or acceptance by determining cytokine levels at the graft site in a mouse model. Ten days after SIS implantation, inflammation was notably reduced and the graft site contained mononuclear cells. The implant was surrounded by increased tissue organization and fibroblastic proliferation, indicating tissue remodeling. On day 28, the inflammatory process resolved, indicating acceptance of the graft. The host response to SIS was like that of syngenetic muscle tissue. The analysis of cytokine RNA from the graft site at the time of mononuclear cell infiltration (seven days after implantation) showed increased levels of IL-4 mRNA, a marker of M2 macrophages, as well as the absence of interferon-γ, a marker of inflammation. The analysis of the antibody isotype also revealed the involvement of the Th2 pathway, which is associated with transplant acceptance.

## 5. M2 Macrophage and Pain Reduction

Inflammatory pain is a severe clinical symptom often experienced by patients with chronic ulcers. Macrophage-produced cytokines strongly influence sensory neurons [33]. Proinflammatory cytokines, such as TNF-α, IL-6, and iNOS, are produced by M1 macrophages and cause inflammatory pain. In contrast, anti-inflammatory cytokines produced by M2 macrophages, such as IL-10 and TGF-β, help mitigate pain. Therefore, a smooth transition from M1 to M2 macrophage also helps resolve the pain associated with inflammation [6]. Previous studies have demonstrated the clinical utility of SIS for controlling pain. Romanelli et al. [34] conducted a randomized trial to compare two ECM biomaterials to treat venous, arterial, and mixed arterial/venous ulcers. Patients were randomized to SIS treatment containing multiple active components of the ECM (*n* = 27) or treatment with a single component of the ECM, hyaluronic acid (Hyaloskin^®^) (*n* = 27). After 16 weeks of treatment, the SIS treatment group showed a significantly faster wound healing rate (*p <* 0.001), less pain (*p <* 0.05), and greater comfort (*p <* 0.01). Hands are very sensitive to external stimuli; therefore, treatment of hand burns usually causes extreme pain to the patient. The dressing of the hand should minimize pain, prevent the formation of scar contracture, and guarantee the greatest extent of functionality possible. Glik et al. [35] compared SIS and Suprathel^®^ (Polymedics Innovations GmbH, Denkendorf, Deutschland) to treat first- and second-degree hand burns. The SIS treatment group showed a decrease in pain level in 70% of patients on the fourth day after surgery. An increase in mature epidermis composed of multiple cell layers was observed within three weeks, and no clinical or histopathological inflammatory reactions were reported. A 5% decrease in rehabilitation risk was observed in the SIS treatment group compared to the Suprathel group.

Furthermore, Celik et al. [36] showed that daily doses of IL-4 applied to the injured peripheral nerve shifted macrophages from the M1 to M2 phenotype, which produced opioid peptides (Met-enkephalin, β-endorphin, and dynorphin) that, upon release, reduce pain in animal models and humans. IL-4 is an anti-inflammatory cytokine that is upregulated by SIS [24]. From this result, IL-4 may be a key mechanism of pain reduction when SIS is used to treat burns and wounds.

## 6. Is Bioactivity Retained?

It is important to determine whether the bioactivity of growth factors is retained for a prolonged period after the implantation of SIS. Hodde et al. [37] investigated the long-term bioactivity of endogenous FGF-2 in SIS using an in vitro bioassay in rat pheochromocytoma (PC12) cells. FGF-2 is present in SIS and plays a key role in wound healing and revascularization of the wound bed. Growth factors are believed to degrade almost immediately after secretion in the ECM. FGF-2 was detected at least 24 months after sterilization and storage at room temperature. Endogenous FGF-2 was maintained in a bioactive form and induced PC12 cell differentiation. Hodde et al. [38] also investigated the impact of several processes, such as peracetic acid (PAA), lyophilization, and ethylene oxide (EO) sterilization, on the composition and three-dimensional matrix structure of SIS. Fibronectin and glycosaminoglycans were retained in SIS after oxidation by PAA and alkylation using EO gas. Significant amounts of FGF-2 were also retained; however, VEGF was susceptible to the effects of PAA and was dramatically reduced after processing. However, in their subsequent study [39], FGF-2 and other factors present in SIS were sufficient to stimulate fibroblasts to secrete their own endogenous VEGF, abrogating the need to retain VEGF in the processed matrix.

Nihsen et al. [40] observed bioactivity in cellular differentiation and proliferation assays, with neurite outgrowth in PC12 cells increasing by approximately 22% relative to negative controls and induction of proliferation in 50.8% of human fibroblasts in an in vitro model. SIS also stimulated VEGF secretion by fibroblasts to a greater extent (422.4 ± 203 pg/mL, mean ± standard deviation) compared to the negative control (mean, 0.0 pg/mL; standard deviation, 1.1 pg/mL; *p <* 0.001) in an in vitro assay. In an in vivo murine model [40], angiogenesis was evaluated using fluorescence microangiography 21 days after implantation, which revealed dense infiltration of blood vessels extending approximately 3 mm from the edge of the implanted disk. SIS contains numerous components that support angiogenesis; however, the role of each component is unknown.

To induce angiogenesis, the presence of a substrate to which endothelial cells can adhere, migrate on, and proliferate within is essential, allowing cells to differentiate into a mature endothelial cell phenotype. Badylak et al. [41] investigated the ability of human microvascular endothelial cells (HMECs) to adhere to SIS and evaluated the role of selected components of SIS in terms of adherence. They exposed the HMECs to hydrated and dehydrated forms of SIS and plastic surfaces coated with one of four components of the SIS: type I collagen, type IV collagen, fibronectin, and laminin. More HMECs adhered to the hydrated forms of SIS than to the dehydrated forms of SIS and plastic dishes containing one SIS component. Electron microscopy analysis revealed that HMECs adhered to hydrated SIS. Although these in vitro findings do not demonstrate the cause of angiogenesis in vivo, they show that human endothelial cells can respond to SIS and that dehydration can cause the denaturation of proteins important for endothelial cell adhesion.

## 7. Does the Matrix-Bound Vesicle Deliver the Bioactive Components to Cells?

Although SIS can change the biochemical balance in a wound from chronic to acute through the improvement of DR, intercellular and intracellular signaling mechanisms are not fully understood. Plausible and logical mechanisms underlying the interaction of ECM bioscaffolds and cells include cues generated by mechanobiology-related cell signaling [42,43], integrin-mediated cell response to the landscape of ECM ligands [44,45], and release of embedded growth factors, cytokinesm, or chemokines [46].

Extracellular vesicles (EVs) are 30–1000 nm (in diameter) membrane vesicles released into the extracellular space by most cell types. EVs are commonly found in cell culture supernatants and in most biologic fluids, such as saliva, plasma, and cerebrospinal fluid [47]. The contents of EVs can regulate diverse physiologic and pathologic processes, such as angiogenesis, immune cell phenotype, cell differentiation and fate, epithelial–mesenchymal transition, and apoptosis [48,49,50]. Huleihel et al. [51] first reported that 10 to 1000 nm EVs, termed matrix-bound nanovesicles (MBVs), are present in three ECM biologic scaffolds, including SIS. MBVs are a distinct class of extracellular vesicle embedded in collagen fibrils within the ECM, which differ from exosomes not only in their location but also in their lack of typical exosomal markers, such as CD63, CD81, CD9, and Hsp70. As with other EVs, MBVs contains bioactive cargoes, including cytokines, carbohydrates, and miRNA, which are transported to cells, tissues, and organs locally and systemically, where they interact through various potential signaling mechanisms to affect cell behavior [52]. MBVs protects cargo from degradation and denaturation during the ECM-scaffold manufacturing process (Figure 3) [51,53]. MBVs can also influence macrophage polarization toward the anti-inflammatory M2-like phenotype and stem cell differentiation. Huleihel et al. [54] showed that specific miRNAs within MBVs can recapitulate the macrophage activation effects of the ECM bioscaffold. The MBV isolated from two source tissues, porcine urinary bladder, and SIS, was enriched in miRNA125b-5p, 143-3p, and 145-5p. Inhibition of these miRNAs within macrophages was associated with a gene and protein expression profile more consistent with a proinflammatory phenotype, rather than an anti-inflammatory/regulatory phenotype. MBVs and its associated miRNA appear to play a significant role in mediating the effects of ECM bioscaffolds on macrophage phenotype. Although most studies on MBVs are in vitro, Merwe et al. [53] showed that intravitreally injected MBVs prevented retinal ganglion cell (RGC) axon degeneration, RGC death, and preserved visual function after severe intraocular pressure-induced ischemia in rats in vivo. Hussey et al. [55] further identified protein signaling molecules associated with MBVs and showed robust expression of IL-33 in porcine SIS MBVs. Therefore, IL-33 may play an important role in identifying MBVs in the near future.

## 8. Conclusions

In this review, we discussed the ability of SIS to alter the wound microenvironment to promote healing by reducing elevated levels of MMPs and shifting the wound state from inflammation to proliferation, as it contains multiple ECM components in a bioactive form. Furthermore, we postulated SIS reduces inflammatory pain by inducing the shift from M1 to M2 macrophages and upregulating IL-4 levels. We also presented the recent discovery of bioactive MBVs within ECM bioscaffolds, including SIS, which can activate macrophages toward an M2-like, pro-remodeling phenotype. Further characterization of MBVs will help to delineate the microenvironment around SIS in wounds. The focus of future studies will be to examine the molecular interactions between SIS and the host (patient) to identify the areas of potential focus for improving the outcomes of patients.

## Figures and Tables

**Figure 1 biomedicines-10-02213-f001:**
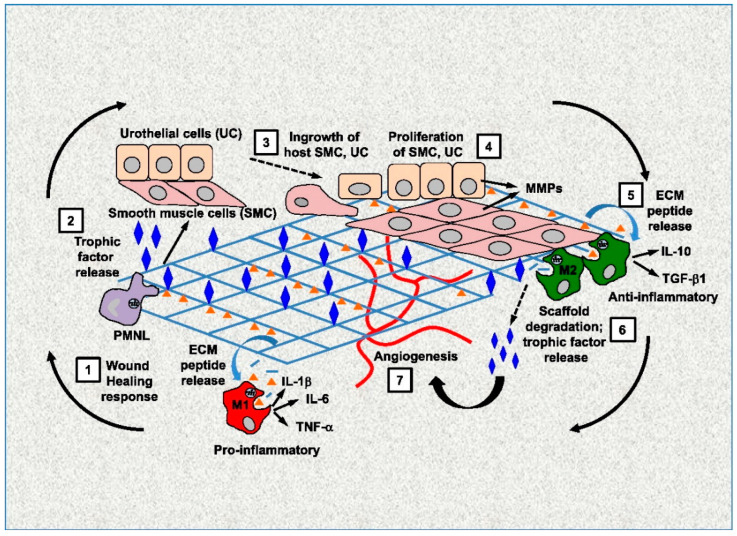
Dynamic reciprocity between cells and ECM. Reprinted with permission from Ref. [4]. Copyright 2022 Elsevier.

**Figure 2 biomedicines-10-02213-f002:**
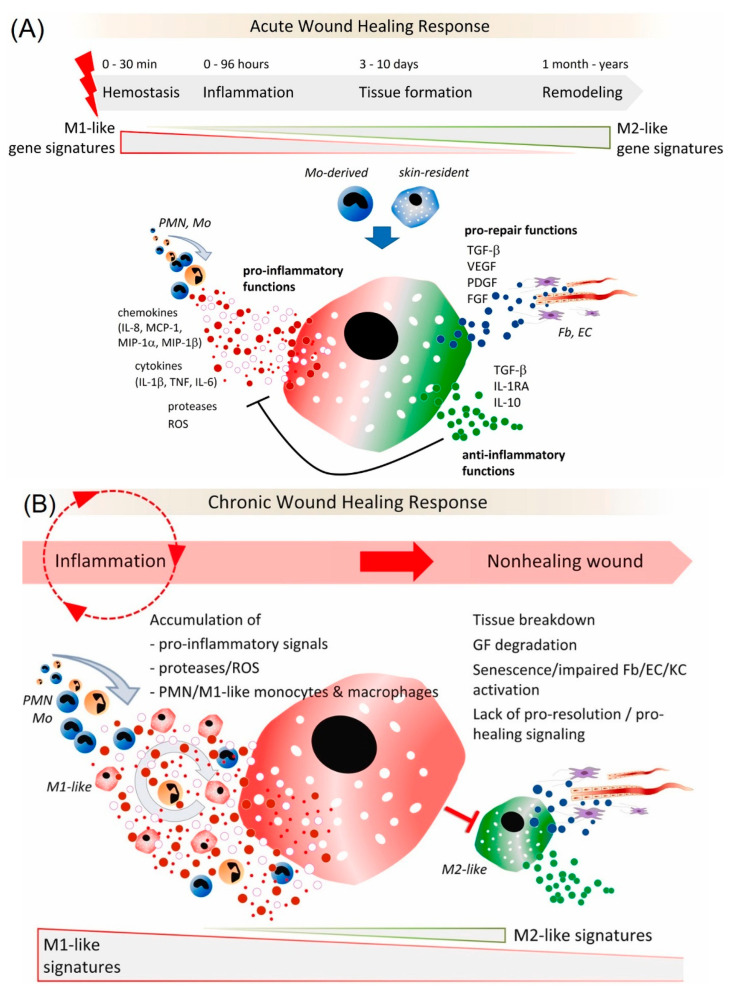
(**A**) Role of macrophages in an acute wound healing response; (**B**) Dysregulated activation of macrophages in chronic wounds. Reprinted from Ref. [26].

**Figure 3 biomedicines-10-02213-f003:**
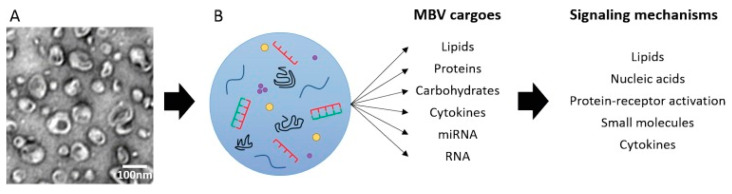
MBV carry diverse cargoes that can signal via multiple mechanisms. Reprinted with permission from Ref. [53]. Copyright 2022 Wolters Kluwer Medknow Publications.

## Data Availability

Data sharing not applicable.

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
