# Peer review of "Porcine Small Intestinal Submucosa Alters the Biochemical Properties of Wound Healing: A Narrative Review"

_biomedicines, 2022, doi:10.3390/biomedicines10092213_

Round 1
Reviewer 1 Report (New Reviewer)
This review discusses the application of SIS for wound healing. Description of clinical studies is a major plus of this article. Some points that can make it stronger are:
on pg 3 line 49 the authors discuss about the potential of SIS to reduce MMPs. Another protease the neutrophil derived elastase (HNE) is also crucial in chronic studies and it should be mentioned for complete coverage. if they are no studies regarding HNE and SIS it should be mentioned.
The article is lacking a future directions paragraph. As to where the state of the art of SIS is in the field of wound healing. What are the current challenges for clinical translations etc.
Author Response
Thank you for your review of our review manuscript. I have revised the manuscript following your advice.
・on pg 3 line 49 the authors discuss about the potential of SIS to reduce MMPs. Another protease the neutrophil derived elastase (HNE) is also crucial in chronic studies and it should be mentioned for complete coverage. if they are no studies regarding HNE and SIS it should be mentioned.
There are no publications to my knowledge on human neutrophil elastase and small intestinal submucosa. However, a recent publication looking at the proteomic profile of ECM reported that SIS contains Leukocyte elastase inhibitor (Wang J Biomed Mater Res 2022). Therefore, I added it on pg4, line 176, ‘Excessive neutrophil activation also leads to an increased production of neutro-phil-derived elastase (HNE) which degrades connective tissue molecules and activates MMPs [24]. A recent publication studying the proteomic profile of ECM reported that SIS contains leukocyte elastase inhibitor [25]. This suggests that SIS may have the potential to reduce elevated levels of HNE.’. And changed the citation number after that.
・The article is lacking a future directions paragraph. As to where the state of the art of SIS is in the field of wound healing. What are the current challenges for clinical translations etc.
I added this sentences in the discussion, pg7, line 349, ‘Further characterization of MBVs will help to delineate the microenvironment around SIS in wounds. The focus of future studies will be to examine the molecular interactions between SIS and the host (patient) to identify the areas of potential focus for improving the outcomes of patients.’.
Reviewer 2 Report (New Reviewer)
In this review paper, authors discussed the ability of SIS to alter the wound microenvironment to promote healing by reducing elevated levels of MMPs and shifting the wound state from inflammation to proliferation, as it contains multiple ECM components in a bioactive form. Furthermore, they postulated SIS reduces inflammatory pain by inducing the shift from M1 to M2 macrophages and upregulating IL-4 levels. They also presented the recent discovery of bioactive MBV within ECM bioscaffolds, including SIS, which can activate macrophages toward an M2-like, pro-remodeling phenotype. This review was well writen and I believe it will attract the interests from the researchers in this field. Readers will gain a lot from this review. However, there is no schematic figures in this review. Before acceptance, I recommend the authors to add at least three schematic figures in this review. The first figure should illustrate the mechanism for SIS promoting constructive remodeling through DR. The second figure should illustrate the mechanism for reducing elevated levels of MMPs. The third figure should illustrate the mechanism for activating macrophages toward an M2-like, pro-remodeling phenotype.
Author Response
Thank you for your recommendation.
I think this is review article, so I should not make it by myself but cite from the papers. Based on your advice, I added three figures.
1) The mechanism for SIS promoting constructive remodeling through DR.
→ I added Figure 1 (Dynamic reciprocity between cells and ECM)
2) The mechanism for activating macrophages toward an M2-like, pro-remodeling phenotype.
→I added Figure 2-1,2 (1 Role of macrophages in an acute wound healing response, 2 Dysregulated activation of macrophages in chronic wounds.)
However, I could not find appropriate figures which illustrate the mechanism for reducing elevated levels of MMPs.
Instead of that, I added Figure 3 (MBV carry diverse cargoes that can signal via multiple mechanisms) to explain the mechanism of MBV.
This manuscript is a resubmission of an earlier submission. The following is a list of the peer review reports and author responses from that submission.
Round 1
Reviewer 1 Report
A review article on small intestinal submucosa from pigs is interesting and this paper provides useful information about its use as bioscaffold and its positive effects in wound healing.
The paper is clearly written and organized. However, there are concerns with the reference, in the introduction, to a specific commercial product. This can be interpreted as a commercial advertisement, it is not good.
There are several bibliographic citations and it is quite impossible that all of them make reference to the same commercial product.
A review should describe the investigations performed on a type of material, and, if of biological origin, on the way it is extracted, prepared and characterized
Another commercial product is cited on page 3 (Aquacel Ag). In this case, a more general description is recommended, avoiding the commercial source.
Therefore, the paper has to be reorganized to be worthy of publication and reach an adequate scientific level, also adding information on the source, way of isolation and purification and characterization of porcine small intestinal submucosa (information should be available in the cited published papers)
In addition, a list of 50 citations is provided, but three of them (14, 15 and 28) cannot be found in the text
Reviewer 2 Report
Manuscript ID: biomedicines-1791639
Title: Porcine small intestinal submucosa alters the biochemical properties of wound healing: a narrative review
The review paper presented the evidence on how small intestinal submucosa (SIS) can shift the biochemical balance in a wound from a chronic to an acute state. The content of this paper is not rigorous enough, especially the parts of the figures and tables to achieve high-quality for review articles.
Other comments are as follows.
1. Can the authors comment on the limitations of each one of the strategies presented?
2. The literature cited in this manuscript is too many old and the overall content is not novel enough to publish in Biomedicines.
3. The content of this paper is not rigorous enough, especially the parts of the figures and tables to achieve high-quality for review articles.
4. Are there any general guidelines drawn-out of the work reported?